# Small and Large Intestine (I): Malabsorption of Nutrients

**DOI:** 10.3390/nu13041254

**Published:** 2021-04-11

**Authors:** Miguel A. Montoro-Huguet, Blanca Belloc, Manuel Domínguez-Cajal

**Affiliations:** 1Departamento de Medicina, Psiquiatría y Dermatología, Facultad de Ciencias de la Salud y del Deporte, University of Zaragoza, 50009 Zaragoza, Spain; 2Unidad de Gastroenterología, Hepatología y Nutrición, Hospital Universitario San Jorge de Huesca, 22004 Huesca, Spain; blanbl@hotmail.com (B.B.); manueldc82@hotmail.com (M.D.-C.); 3Aragonese Institute of Health Sciences (IACS), 50009 Zaragoza, Spain

**Keywords:** malabsorption, maldigestion, micronutrients

## Abstract

Numerous disorders can alter the physiological mechanisms that guarantee proper digestion and absorption of nutrients (macro- and micronutrients), leading to a wide variety of symptoms and nutritional consequences. Malabsorption can be caused by many diseases of the small intestine, as well as by diseases of the pancreas, liver, biliary tract, and stomach. This article provides an overview of pathophysiologic mechanisms that lead to symptoms or complications of maldigestion (defined as the defective intraluminal hydrolysis of nutrients) or malabsorption (defined as defective mucosal absorption), as well as its clinical consequences, including both gastrointestinal symptoms and extraintestinal manifestations and/or laboratory abnormalities. The normal uptake of nutrients, vitamins, and minerals by the gastrointestinal tract (GI) requires several steps, each of which can be compromised in disease. This article will first describe the mechanisms that lead to poor assimilation of nutrients, and secondly discuss the symptoms and nutritional consequences of each specific disorder. The clinician must be aware that many malabsorptive disorders are manifested by subtle disorders, even without gastrointestinal symptoms (for example, anemia, osteoporosis, or infertility in celiac disease), so the index of suspicion must be high to recognize the underlying diseases in time.

## 1. Introduction

Nutrient digestion and absorption are necessary for the survival of living organisms and have evolved into the complex and specific tasks of the gastrointestinal (GI) system. Thus, in healthy conditions, the GI tract will work properly to use nutrients, provide energy, and release wastes. However, numerous disorders can alter the physiological mechanisms that guarantee proper digestion and absorption of nutrients (macro- and micronutrients), leading to a wide variety of symptoms and nutritional consequences.

Malabsorption can be caused by many diseases of the small intestine, as well as by diseases of the pancreas, liver, biliary tract, and stomach. This article provides an overview of pathophysiologic mechanisms that lead to symptoms or complications of maldigestion (defined as the defective intraluminal hydrolysis of nutrients) or malabsorption (defined as defective mucosal absorption), as well as its clinical consequences, including both gastrointestinal symptoms and extraintestinal manifestations and/or laboratory abnormalities [1]. The clinician must be aware that many malabsorptive disorders are manifested by subtle disorders, even without gastrointestinal symptoms (for example, anemia, osteoporosis, or infertility in celiac disease), so the index of suspicion must be high to recognize the underlying diseases in time.

The normal uptake of nutrients, vitamins, and minerals by the gastrointestinal tract (GI) requires several steps, each of which can be compromised in disease, and there are excellent reviews about it [2]. This article will first describe the mechanisms that lead to poor assimilation of nutrients and, secondly, discuss the symptoms and nutritional consequences of each specific disorder. Due to its global vision, this manuscript seeks to be useful both to “registered dietitian-nutritionists” (RDN) and to family doctors, gastroenterologists, internists, and surgeons. Regarding the first (RDN), physicians need to have a reliable ally for their patients’ nutritional management, but this is only possible if the RDN demonstrates robust knowledge about the causes and consequences of diseases on nutritional status.

## 2. Causes of Nutrient Malabsorption

Next, we will describe the conditions causing malabsorption, categorized by the phase of the absorption that is impaired: the luminal phase, the mucosal (absorptive) phase, and postabsorptive processing.

### 2.1. Luminal Phase

The luminal phase primarily affects the digestion of some macronutrients (fats and proteins) and also of micronutrients whose absorption depends on their adequate assimilation, especially fat-soluble vitamins (vitamins A, D, K, and E), as well as minerals such as calcium, dependent on the correct absorption of fats, and others such as vitamin B12, dependent on adequate secretion of intrinsic factor and pancreatic proteases (see Section 1).

Most dietary lipids are absorbed in the proximal two-thirds of the jejunum. In normal children and adults, more than 94% dietary fat is absorbed. As a result, in a diet containing 100 g of fat per day, the presence of >6 g of fecal fat in a 24 h collection indicates fat malabsorption. The digestion of lipids can begin in the mouth, with lingual lipase produced by glands in the tongue, and continues in the stomach, with lingual lipase and gastric lipase produced by chief cells. However, in adult humans, most fat arrives in the duodenum intact, as only ~15% of fat digestion occurs by the time the food leaves the stomach [2,3].

Emulsification of dietary fat is facilitated by cooking the food, continues with chewing, and finishes in the stomach with churning and peristalsis [2]. The emulsion droplets arriving from the stomach contain almost all the dietary triglycerides and diglycerides in their cores and are covered by polar lipids, phospholipids, fatty acids, cholesterol, triglycerides, denatured dietary proteins, dietary oligosaccharides, and bile salts in the duodenum.

The lipid emulsion is exposed to pancreatic lipase, which enzymatically degrades each triglyceride molecule into a 2-monoglyceride and two fatty acids. Optimizing the activity of pancreatic lipase depends on the presence of another enzyme, colipase, which is secreted as procolipase and activated by trypsin [4]. Pancreatic lipase hydrolyzes fatty acids at positions 1 and 3 of the glycerol moiety and produces free fatty acids and a 2-monoglyceride (also known as a monoacylglyceride).

Some conditions required for the adequate digestion of fats include

the adequate hydrolysis of the substrate, dependent on a sufficient concentration and mixture of pancreatic enzymes;the correct solubilization of fats in water (emulsion), which depends, in turn, on the adequate synthesis, transport, and concentration of bile salts in the intestinal lumen;the adequate availability of specific nutrients, especially cobalamin (vitamin B12), also dependent on a sufficient secretion of intrinsic factor in the stomach.

Protein digestion begins in the stomach by the action of gastric pepsins, which are released as proenzymes (pepsinogen 1 and 2) and undergo autoactivation at low pH. The observation that patients who are achlorhydric or have rapid gastric emptying can still digest proteins suggests that proteolysis within the stomach is not essential for protein digestion. In the duodenum, several proteases act together to digest proteins into amino acids, or dipeptides and tripeptides. Enterokinase, which is released from the microvillus membrane of duodenal absorptive cells by the action of bile salts, converts trypsinogen to trypsin, which then catalyzes the conversion of all other pancreatic proteases to their active forms.

Therefore, the digestion of proteins depends largely on the following:adequate intragastric conversion of dietary proteins into amino acids, which play a role in releasing cholecystokinin (CCK) from duodenal and jejunal endocrine epithelial cells and releasing pancreatic enzymes to the duodenojejunal lumen;an adequate concentration and mixture of pancreatic ferments with the ingested nutrients;an adequate concentration of bile salts in the intestinal lumen (responsible for the release of enterokinase).

Some diseases and mechanisms that can alter the absorption of fats and proteins in the luminal phase of digestion are described below.

#### 2.1.1. Substrate Hydrolysis

##### Digestive Enzyme Deficiency

The pancreas secretes approximately 1.5 L of enzyme-rich fluid every day for the digestion of fats, starch, and protein. Stimulation of the pancreas after a meal increases the flow of water, bicarbonate (to neutralize gastric chyme for optimal digestion), and a large volume of alkaline, enzyme-rich fluid. All this is controlled by hormonal and neuronal mechanisms, with the main regulatory hormones being secretin and cholecystokinin (CCK).

The impaired digestion and absorption of dietary fat occurs when pancreatic enzyme secretion (including that of lipases, colipases, and esterases) and/or activity is impaired. In this context, there is a decrease in the luminal hydrolysis of the dietary fat. Chronic pancreatitis, cystic fibrosis, pancreatic duct obstruction by pancreatic and ampullary tumors, and pancreatic resection are the most common causes of pancreatic insufficiency. Similarly, defective proteolysis also occurs with exocrine pancreatic insufficiency, as well as by inborn errors in the synthesis of proteolytic enzymes (trypsinogen deficiency) or by defective activation of pancreatic proenzymes resulting from congenital deficiency of intestinal enterokinase.

Chronic pancreatitis is a complex chronic inflammatory condition that results in pancreas fibrosis and calcification, with subsequent exocrine and endocrine insufficiency. Alcohol and nicotine are the most prevalent risk factors (see manuscript # 4 of this special issue). When alcohol abuse is present in its etiology, the nutritional consequences of pancreatic exocrine insufficiency must be added to those induced by the alcohol itself [5].

Cystic fibrosis (CF) is the most common worldwide, life-shortening multisystem hereditary disease, with an autosomal recessive inheritance pattern caused by mutations in the cystic fibrosis transmembrane conductance regulator (CFTR) gene. Approximately 80% of patients with cystic fibrosis also develop progressive pancreatic damage due to the blockage of ductules resulting from inspissated pancreatic secretion [6].

In cancer of the pancreatic head region, exocrine insufficiency is a well-known complication, leading to steatorrhea, weight loss, and malnutrition. Its presence is frequently overlooked, however, because the primary attention is focused on cancer treatment [7].

Primary ampullary neoplasms have their origin in the ampulla of Vater, an anatomical structure where the common bile duct and the pancreatic duct join together as a common channel. They represent <0.5% of all gastrointestinal cancers and approximately 7% of all periampullary cancers [8].

Pancreatic resection. Exocrine insufficiency frequently develops in patients with pancreatic cancer owing to tumor ingrowth and pancreatic duct obstruction. Surgery might restore this function by removing the primary disease and restoring duct patency, but it may also have the opposite effect because of the resection of functional parenchyma and anatomical changes [9].

Trypsinogen deficiency. This rare disease remains difficult to diagnose, highlighting the importance of considering pancreatic etiologies in the workup of failure to thrive with poor weight gain despite adequate caloric intake forage. Clearly, we have made progress in our ability to obtain pancreatic fluid samples through endoscopy, and there is increased availability of commercial laboratories to run the pancreatic enzymes assays. A low trypsin level is consistent with exocrine pancreatic insufficiency [10].

Congenital deficiency of intestinal enterokinase. Enterokinase is a glycoprotein present in the duodenal and jejunal mucosa and is now designated as enteropeptidase [11]. Enterokinase converts trypsinogen to trypsin in the duodenal lumen. Congenital enterokinase deficiency is a distinct clinical entity characterized by diarrhea, failure to thrive, hypoproteinemia, and edema [12]. Acquired enterokinase deficiency may occur in some diffuse small bowel diseases. In fact, steatorrhea of celiac spruce may be due partly to the fact that a deficiency of secretin and cholecystokinin may interfere with the action of enterokinase [13].

Shwachman‒Diamond syndrome is a rare disease associated with an increased risk of myelodysplasia and leukemia, as well as pancreatic exocrine dysfunction, neutropenia, and bone abnormalities, and genetic testing is helpful for diagnosis [14]. Other rare causes of exocrine pancreatic insufficiency include hereditary hemochromatosis, which results in progressive iron deposition in the pancreas.

##### Digestive Enzyme Inactivation

The entry of gastric hydrogen ions into the duodenum stimulates the release of secretin, which enhances pancreatic bicarbonate secretion. This raises the intraluminal pH to approximately 6.5, which is optimal for fat digestion. Diseases that substantially decrease duodenal pH, such as Zollinger‒Ellison syndrome (ZES), can selectively inhibit fat absorption.

ZES is a group of symptoms comprising severe peptic ulcer disease, gastroesophageal reflux disease (GERD), and chronic diarrhea caused by a gastrin-secreting tumor of the duodenum or pancreas (gastrinoma triangle) that results in increased stimulation of the acid-secreting cells of the stomach.

In the ZES, the rate of gastric acid secretion exceeds the neutralizing capacity of pancreatic bicarbonate secretion, resulting in an exceptionally low pH of the intestinal contents. The low pH inactivates pancreatic digestive enzymes, interfering with the emulsification of fat by bile acids, and damaging intestinal epithelial cells and villi. Maldigestion and malabsorption both result in steatorrhea [15].

##### Dyssynchrony of Enzyme Release, Inadequate Mixing

For sufficient digestion and absorption of lipids, dietary fat must adequately mix digestive secretions. Gastric resections or GI motility disorders that result in rapid gastric emptying or rapid intestinal transit, such as autonomic neuropathy due to diabetes mellitus or amyloidosis, can cause fat malabsorption consequent to impaired GU mixing of dietary fat [1]. The number of surgeries done to correct obesity (e.g., Roux-en-Y gastric bypass) continues to rise, underscoring the need to appreciate nutrient deficiencies that commonly accompany the resulting weight loss. This surgical procedure produces mild fat malabsorption of a multifactorial nature, including an inadequately mix digestive secretions [16,17]. A more detailed description of the nutritional consequences associated with bariatric surgery has been described in Section 1.

#### 2.1.2. Fat Solubilization

The presence of fat in the duodenum leads to the contraction of the gallbladder, with the relaxation of the hepatopancreatic sphincter, to release bile in the upper part of the duodenum. The emulsion is stabilized by preventing the dispersed lipid particles from coalescing again by coating them with bile salts, phospholipids, and cholesterol. Since digestive lipases have adapted to being more efficient at oil‒water interfaces, turning dietary fat into an emulsion of fine oil droplets that enhances the action of lipases [2]. Correct solubilization of fats in water (emulsion) depends on the adequate synthesis, transport, and concentration of bile salts in the intestinal lumen. Fat malabsorption resulting from decreased formation of micelles occurs if the luminal concentrations of conjugated bile acids are lower than the critical concentration required for forming micelles [18]. The following are representative diseases that cause luminal bile acid deficiency by different mechanisms.

##### Decreased Synthesis and/or Secretion of Conjugated Bile Acids

Cirrhosis represents a late stage of progressive hepatic fibrosis characterized by distortion of the hepatic architecture and the formation of regenerative nodules. Fat malabsorption is frequent in cirrhotic patients, particularly when malnourished, and does not depend on the presence of mucosal intestinal damage, but rather from a failure in the synthesis and excretion of bile salts, especially in forms of primary biliary cholangitis where cholestasis is a prominent feature [19,20]. Nutritional abnormalities in patients with liver cirrhosis are discussed in Section 4.

Other causes of cholestasis that decrease the concentration of bile salts in the duodenal lumen include malignant biliary tract obstruction, primary sclerosing cholangitis, prolonged drug-induced cholestasis, intrahepatic cholestasis of pregnancy, chronic viral hepatitis, and inherited cholestasis syndromes (e.g., progressive familial intrahepatic cholestasis and benign recurrent intrahepatic cholestasis). Figure 1 shows the case of a patient with steatorrhea due to chronic pancreatitis and cholestasis due to common bile duct involvement. External biliary fistulae, often caused by iatrogenic injury after operations, invasive procedures, or trauma involving the biliary tract, involve a loss of bile salts, which has a direct impact on their pool in the intestinal lumen [21,22]. Inborn errors of bile acid synthesis are rare genetic disorders that can present as neonatal cholestasis, neurologic disease, or fat-soluble-vitamin deficiencies. Failure to diagnose any of these conditions can result in liver failure or progressive chronic liver disease [23]. A defective release of CCK is seen in some conditions such as celiac disease and markedly increases susceptibility to cholesterol gallstones via a mechanism involving dysmotility of both the gallbladder and the small intestine [24].

##### Bile Salt Deconjugation

Small intestinal bacterial overgrowth (SIBO) is a condition in which colonic bacteria are seen in excess in the small intestine. Luminal deconjugation of bile acids by florid small bowel bacterial overgrowth defunctionalizes the bile acids and can also result in fat malabsorption. Bacterial deconjugation also leads to the production of lithocholic acid, which may be toxic to intestinal epithelium, resulting in impaired absorption of fat and other nutrients. Hydroxylated fatty acids (and free bile acids) also stimulate the secretion of water and electrolytes, leading to diarrhea. Some mechanisms involved in the pathogenesis of SIBO are the following:Anatomic disorders, such as gastric bypass for the treatment of obesity, tumors of the small bowel, adhesions from previous surgery or strictures due to radiation or inflammatory bowel disease, blind intestinal loops, and small intestinal diverticulosis. In all these circumstances, intestinal stasis occurs, favoring the growth of bacteria such as *Escherichia coli*, *Klebsiella* spp, and *Aeromonas*, among others.Functional and motility disorders. Narcotic use, intestinal pseudo-obstruction, diabetes, and even irritable bowel syndrome [25,26], and other factors such as scleroderma lead to a lack of activity of phase III of the migrating motor complex, and a consequent failure to cleanse the small bowel.Immune disorders. Intestinal immunity is important for maintaining the correct microbial composition in the small intestine. Combined variable immunodeficiency, IgA deficiency, acquired immunodeficiency (e.g., HIV), and even immunosenescence (often aggravated by using antisecretory drugs) [27] are associated with an increased risk of SIBO.Metabolic and systemic disorders. Diabetes mellitus and Parkinson’s disease can cause SIBO due to intestinal neuropathy. Disorders such as pancreatic insufficiency and cirrhosis can predispose to SIBO by changing the quantity and composition of these digestive enzymes or bile, thereby allowing microbes to grow. SIBO has long been regarded as a potential complication of celiac disease and one of the causes of failure to respond to gluten withdrawal. Overall, SIBO is responsible for approximately 10‒20% of nonresponsive celiac disease [28,29,30], though some studies suggest that SIBO may coexist with other disorders associated with nonresponsive CD [30]. SIBO has been frequently documented in association with alcoholic or nonalcoholic liver disease [31,32], and is more likely in patients with decompensated cirrhosis of the liver with portal hypertension, ascites, and jaundice. Intestinal bacteria are clearly fundamental to overt hepatic encephalopathy and even minimal hepatic encephalopathy.

##### Increased Bile Salt Loss

Small bowel disease, or resection, of greater than 100 cm of terminal ileum commonly results in severe impairment of the enterohepatic circulation of bile salts, such that the liver’s ability to upregulate de novo bile acid synthesis is inadequate to meet normal physiological needs for bile production, resulting in fat malabsorption. In addition, the increased passage of bile acids into the colon may induce a colonic secretomotor diarrhea to stimulate water and electrolyte secretion in the colon (which is called “cholerrheic diarrhea” or “choleretic enteropathy”) [33,34,35].

#### 2.1.3. Luminal Availability of Specific Nutrients

##### Diminished Gastric Acid

Iron is absorbed in the upper gastrointestinal tract; the duodenum is the site of maximal absorption. Heme dietary sources (fish, poultry, and meat) have higher bioavailability than do nonheme (vegetable) sources (30 versus <10%). In addition, intraluminal factors can affect absorption. Iron in food is prominently ferric (Fe^3+^), which is poorly soluble above a pH of 3 and is therefore poorly absorbed. In comparison, ferrous iron (Fe^2+^) is more soluble, even at the pH of 7 to 8 seen in the duodenum. As a result, it is more easily absorbed. Thus, the conversion of (Fe^3+^) to ferrous salts (Fe^2+^) by the acidic pH of the stomach is an important step in its absorption. Certain medical conditions that cause gastric acid hyposecretion, such as chronic atrophic gastritis or long-term, potent acid inhibition caused by proton pump inhibitors, may interfere with the normal uptake of dietary iron and be the cause of iron deficiency, with or without anemia [36].

##### Diminished Intrinsic Factor

Chlorhydropeptic and intrinsic factor secretion by parietal cells is essential for the absorption of vitamin B12. Vitamin B12 in foods is protein-bound, and it is dissociated in the acid milieu of the stomach with the help of pepsin. Additional vitamin B12-binding proteins known as R-binders are secreted in the saliva and bind to vitamin B12 in the stomach. Gastric parietal cells produce intrinsic factor and pancreatic proteases secreted into the higher pH duodenum cleave off the R-binders, allowing vitamin B12 to bind to the intrinsic factor. Those conditions result in a profound inhibition of gastric acid, and intrinsic factor secretion leads to a state of selective malabsorption of vitamin B12. Such is the case with autoimmune atrophic chronic gastritis, a cause of pernicious anemia [37].

##### Bacterial Consumption of Nutrients

Some typical laboratory findings of SIBO are elevated folate and, less commonly, vitamin B12 deficiency, especially if intake is low and/or if stores are borderline. They often have one or more predisposing factors, such as gastric surgery; atrophic gastritis; chronic *H. pylori* infection; achlorhydria from chronic use of antacids, H2 receptor blockers, or proton pump inhibitors; chronic excess alcohol use; pancreatic insufficiency; or antibiotic use. Some of these conditions affect the elderly population more frequently. In fact, mild and/or subclinical vitamin B12 deficiency has been documented at relatively high frequency in older adults, with various observational studies describing a prevalence of approximately 5 to 20% depending on the population studied and the laboratory criteria used to define deficiency [38]. In SIBO, several factors conspire to cause B12 deficiency: the consumption of cobalamin by anaerobes, malabsorption of the vitamin due to competitive binding with cobalamin from bacterially generated metabolites of cobalamin at the ileal receptor, and, in more severe overgrowth, mucosal injury involving the binding site [39]. Bacterial synthesis of folic acid may result in the unusual combination of high folate and low B12 levels.

### 2.2. Mucosal (Absorptive) Phase

Dietary carbohydrates mainly include sucrose and a variety of plant starches, which are composed of different α-linked sugars. Sugar transporters in the intestine are only capable of transporting monosaccharides. Therefore, the accessibility of higher carbohydrates to the human body as energy sources necessitates their hydrolysis to simple monosaccharides.

Initial digestion of the complex carbohydrates begins with salivary α-amylase while still in the mouth. Salivary α-amylase is deactivated by acid pH, so it remains active in the stomach only if it is protected from stomach acid. Inside the small intestine, pancreatic juice enters the lumen through the hepatopancreatic sphincter (sphincter of Oddi), and its high bicarbonate concentration begins to neutralize gastric acid. Since di-, tri-, and oligosaccharides result from the hydrolysis of starch by α-amylase, additional digestion is required before the absorption of the monosaccharide breakdown products of starch can occur. These starch hydrolysis products must be further broken down by the disaccharidases (maltase, sucrase-isomaltase complex, and β-glycosidase complex, which includes lactase and glucosyl-ceramidase), found as membrane-spanning enzymes in the plasma membranes of the brush borders of intestinal epithelial cells (enterocytes) [2,40].

Osmotic diarrhea and abdominal pain in humans are often associated with carbohydrate malabsorption in the small intestine due to the loss of function of microvillar disaccharidases.

#### 2.2.1. Brush Border Hydrolysis

##### Acquired Disaccharidase Defect (Lactase Deficiency)

Lactose is a disaccharide that is present in many dairy products, composed of galactose linked to glucose via a β-1→4 glucosidic bond. Lactose is hydrolyzed by β-galactosidase (lactase) bound to the small intestine brush border membrane, and the monosaccharides glucose and galactose are both actively absorbed in the small intestine. Lactose intolerance is one of the most common forms of food intolerance and occurs when lactase activity is reduced in the brush border of the small bowel mucosa [40]. There are three types of lactose intolerance:congenital lactase deficiency (CLD): an extremely rare autosomal recessive disease characterized by absent or reduced enzymatic activity from birth;primary lactose intolerance or adult-type lactase deficiency: a common autosomal recessive condition resulting from a developmentally regulated change of the lactase gene expression;secondary lactase deficiency: a transient condition deriving from intestinal damage secondary to several diseases such as infections, celiac disease, food allergy, small bowel bacterial overgrowth, Crohn’s disease, or radiation/chemotherapy-induced enteritis.

Certainly, all pathological conditions that cause small intestine damage can induce a reduction in lactase expression, determining a secondary and transient lactase deficiency. The treatment of the underlying disorder may restore lactase levels and improve signs and symptoms, though it can take time. Cow’s milk allergy (CMA) can cause severe enteropathy with secondary lactase deficiency. In these patients, there may be an overlap of gastrointestinal symptoms due to CMA and lactose intolerance [41,42,43].

Symptoms due to lactose intolerance are described later.

##### Congenital Disaccharidase Defect (Sucrase-Isomaltase Deficiency)

Congenital sucrase-isomaltase deficiency (CSID) is a genetically determined primary defect of sucrase-isomaltase (SI) that is associated with carbohydrate malabsorption [44,45,46]. CSID is elicited by a single-nucleotide polymorphism in the SI gene, leading to single amino acid exchanges in the protein. Based on the mutation type, the combined effects of two mutations or of one mutation with the wild-type protein can set the course of the disease and generate mild to severe symptoms. The CSID symptoms are like those of other intestinal diseases, which often leads to misdiagnosis or late diagnosis after childhood. The major secondary or induced SI deficiencies can be categorized into three main groups: (a) those that are induced by physical injuries to the intestine and disrupt the intestinal epithelium, (b) those that are caused via inhibitory function of some dietary components or therapeutic agents on the function of SI, and (c) those that are connected to infections or autoimmune disorders, including celiac disease.

##### Trehalase Deficiency

Trehalose is a disaccharide, the main dietary source being mushrooms. It has been approved as an additive in the preparation of dried food. Isolated intestinal trehalase de-ficiency is found in 8% of Greenlanders but is rare elsewhere. Activity is significantly reduced in untreated celiac disease and recovers with treatment with a gluten-free diet. There is no place for routine determination of trehalase activity in the population, and there should be no concern over the introduction of trehalose-containing dried foods [47].

##### Fructose Intolerance

Dietary fructose is consumed in two forms (as a monosaccharide and as a disaccharide) since it is a component of the disaccharide sucrose (glucose‒fructose). The ability to absorb fructose depends not only on the amount of fructose consumed, but also on the presence of other sugars ingested with it. Concomitant ingestion of glucose, galactose, and some amino acids increases fructose absorption, while sorbitol decreases it [48,49]. These observations have implications for dietary recommendations in patients in whom fructose malabsorption is suspected, since foods that contain fructose may be well tolerated if they also contain glucose but may be less well tolerated if they also contain sorbitol (although sorbitol itself can cause diarrhea).

With the rising use of fructose as a sweetener in commercially prepared foods, often as high-fructose corn syrup, concern about fructose intolerance has emerged, although the clinical syndrome remains a matter of controversy. Some individuals seem to develop symptoms of carbohydrate malabsorption when sufficiently large quantities of fructose are consumed, particularly if the fructose is in a form other than as one of the mono-saccharide components of sucrose.

Unabsorbed fructose provides a substrate for rapid bacterial fermentation (causing gas), and may have other physiological consequences, including an increased osmotic load and the alteration of gastrointestinal motility (which may cause diarrhea), and a change in the profile of the bacterial flora. Fructose-rich foods include honey; sweetened juices; jams; cocoa and chocolate; “sugar-free” soft drinks; candy; sweets; some fruits such as apples, pears, peaches, plums, prunes, and cherries; and vegetables such as cabbage, cauliflower, broccoli, Brussels sprouts, lettuce, and legumes. Patients with fructose in-tolerance should reduce their consumption of these foods when intolerance is a notable clinical problem [50].

Primary fructose malabsorption results from defects in the fructose transporter (GLUT-5).

### 2.3. Epithelial Transport

Once monosaccharides result from the digestion of carbohydrates by α-amylase and the brush-border membrane enzymes, the monosaccharides are taken up by the enterocytes via specific transport proteins that facilitate the transport of the D-isomers (but not L-isomers) of hexoses, by either active or passive transport processes [2].

On the other hand, following pancreatic enzyme digestion, amino acids, dipeptides, and tripeptides can be absorbed through highly efficient sodium-dependent amino acid co-transporters at the brush border membrane; this step is passive but is called secondary active transport since the energy is indirectly provided by the sodium‒potassium ATPase pump. Different classes of amino acid transporters exist on the brush border. Transporters for dipeptides and tripeptides are also distinct from those responsible for free amino acid transport. Additional peptidases are located within the brush border membrane and the cytoplasm of absorptive cells [51]. Several loss-of-function mutations in amino acid transporters have been characterized based on the expression of similar transporters in small intestinal epithelial cells and renal proximal tubule cells (see below).

Regarding lipids, fatty acid translocase (FAT; known as FAT/CD36) appears to play a key role in the uptake of long-chain fatty acids in the small intestine, with higher levels found in the proximal intestinal mucosa. FAT/CD36 is highly expressed and upregulated in the presence of dietary fat, genetic obesity, and diabetes mellitus [52]. In addition, fatty acid transport proteins (FATP2–FATP4) are expressed in the small intestine [53] and are promising candidates for cellular long-chain fatty acid transporters that facilitate the uptake of fatty acids into the enterocytes [54]. The breakdown products of triglyceride hydrolysis, which enter the intestinal epithelial cells across the apical membranes, cross the cytoplasm to the smooth endoplasmic reticulum to be reconstituted into complex lipids. Specific fatty acid-binding proteins (FABPs) carry cytoplasmic fatty acids and monoglycerides to the intracellular sites, where several enzymes reassemble the fatty acids and monoacylglycerides to reconstitute triglycerides. Triglycerides, cholesterol esters, phospholipid, and apoprotein B48 form an aggregate, which is then transferred to the Golgi body for further processing into a fully mature chylomicron, which then binds to the basolateral membrane and is transported to join the circulation via the thoracic duct lymph [55].

#### 2.3.1. Nutrient-Specific Defects in Transport (Hartnup Disease and Cystinuria)

Two autosomal recessive disorders of amino acid transport across the apical membrane have provided extensive insights into the absorption of amino acids and oligopeptides.

Hartnup disease, an autosomal recessive defect named after an English family, was described in 1956. This disease results from the impaired transport of neutral amino acids across epithelial cells in renal proximal tubules and intestinal mucosa due to mutations in the SLC6A19 gene, which encodes for a sodium-dependent neutral amino acid transporter that is primarily expressed in the kidney and intestine. Symptoms include transient manifestations of pellagra (rashes), cerebellar ataxia, and psychosis [56].

Cystinuria is aminoaciduria due to the impairment of transport of cystine and dibasic amino acids (arginine, ornithine, and lysine) in the apical membrane of the intestinal epithelium and proximal renal tubule. The result is an absence of cystine reabsorption in the renal tubule, producing an excess of cystine in urine and stone formation [57].

Neither of these conditions (Hartnup disease and cystinuria) involves the oligopeptide cotransporter, and patients rarely exhibit protein deficiencies.

#### 2.3.2. Global Defects in Transport

Malabsorption most often occurs from diseases leading to a reduced intestinal absorptive area, such as celiac disease, and other diseases causing enteropathy and varying degrees of villus atrophy.

##### Celiac Disease

Celiac disease (CeD) is an autoimmune disorder characterized by a systemic response to dietary gluten in genetically predisposed individuals, which has clinical manifestations of small bowel enteropathy associated with gastrointestinal as well as non-gastrointestinal symptoms [58,59]. In most patients, it is associated with the expression of HLA haplotypes DQ2 and DQ8 [58,60]. However, the presence of these alleles does not represent CeD in the absence of an immune response, which can be triggered by the binding of deamidated gliadin peptides (DAPs) to antigen-presenting cells. CeD affects the mucosa of the small intestine, with degrees of injury that can vary considerably in severity and extent. Consequently, the nutritional effects of CeD vary enormously from patient to patient.

In CeD, there is a compromised absorptive capacity, but there is also evidence that secretin and cholecystokinin release in response to a meal is impaired, thus diminishing the delivery of bile and pancreatic secretions to the intestinal lumen and possibly compromising intraluminal digestion. In CeD, therefore, at least two phases involved in the assimilation of nutrients (the luminal phase and the mucosal (absorptive) phase) may be impaired. As patients with CeD get older, they tend to present with complaints not directly referable to the GI tract. These extraintestinal symptoms and clinical findings often result from nutrient malabsorption and can involve virtually all organ systems (e.g., anemia, osteoporosis, and neurological manifestations) (see below) [61,62,63,64]. Figure 2 shows the case of a patient with “idiopathic” osteoporosis that turned out to be a seronegative celiac disease without abdominal symptoms (non-classical pattern).

Refractory celiac disease (types I and II) (RCDI) is defined by persistent villous atrophy despite a strict GFD longer than one year that leads to concomitant symptoms such as diarrhea, steatorrhea, abdominal pain, weight loss, and/or nutritional deficiencies. Approximately half of RCD-II patients develop T-cell lymphoma associated with enteropathy (EATL) within five years. EATL are often located extra-intestinally, which agrees with the finding that IELs are able to disseminate and can be found in the epithelium of the stomach and colon, as well as the bronchi and the skin. The five-year survival in RCDII patients is poor (44–58%), which, in addition to severe malabsorption, is mainly due to the development of an EATL [65,66,67].

An additional concern in CeD is that many gluten-free foods not only are deficient in several nutrients [68,69], but also, due to their composition, are higher in lipids, salt, and sugar [70,71]. In fact, the increased lipid content of most gluten-free foods, especially breads and flour, can further add to cardiovascular disease risk. Regarding carbohydrates, CeD patients are at a higher obesity risk due to the high glycemic index and glycemic load of a gluten-free diet [72].

##### Other Diseases Causing Enteropathy

The clinician must be aware that there are a wide variety of digestive and systemic diseases that share histological features typical of CeD, including autoimmune, neoplastic, inflammatory, infiltrative, infectious, and ischemic conditions, as well as medication-related ones [73]. All of these can be the cause of nutritional deficiencies.


*Immunomediated Enteropathy*


Crohn’s disease (CrD) is an inflammatory disease that can affect any part of the intestinal tract, with the small intestine being affected in 80% of patients. The most common site of small intestinal inflammation is the terminal ileum, while the proximal small intestine is affected in only 5% of CrD patients. Such cases can present with malabsorption, most often affecting micronutrients (iron, calcium, vitamin D3, vitamin B12, calcium, copper, zinc, and selenium). Some keys to diagnose the disease are the elevation of acute-phase reactants (erythrocyte sedimentation rate, C-reactive protein, and fecal calprotectin) that are not common in celiac disease.

Due to transmural inflammation, patients with CrD will often display symptoms resulting from intestinal ulceration and stenosis, as well as from a fistula or abscess elsewhere in the abdomen or perianal region. Malnutrition associated with inflammatory bowel disease has a multifactorial origin [74,75] and includes the following:anorexia from underlying disease,chronic nausea and/or vomiting,oropharyngeal ulceration,malabsorption,gastrointestinal tract obstruction,fistulizing disease,increased energy expenditure,hypercatabolism, associated with sepsis and systemic inflammatory responses syndrome,extensive surgical resection of the small bowel,small intestinal bacterial overgrowth,medication side effects, andNPO for diagnostic workup.

Autoimmune or immune-mediated enteropathy (AIE) is an immune disorder of the gut seldom found in adults, caused by excessive reactions of the intestinal immune system towards nonpathogenic molecules. The disease is characterized by the presence of chronic diarrhea with malabsorption, often severe, weight loss and vitamin deficiencies, histological findings of partial or complete blunting of small intestinal villi, increased apoptotic bodies, and deep crypt lymphocytosis with minimal intraepithelial lymphocytosis. The presence of circulating autoantibodies provides major diagnostic support, but its absence does not exclude AIE [76,77]. It is typically associated with other autoimmune conditions and mesenteric lymphadenopathy. Parenteral feeding and immunosuppressive therapy are needed in severe cases.

Common variable immunodeficiency syndrome (CVID) is characterized by an impaired antibody response that leads to the insufficient production of immunoglobulins (prevalence: 1 in 25,000‒50,000). About half of patients with CVID experience chronic diarrhea [74,75]. It is believed that gastrointestinal infections are the cause of diarrhea. Infection with *Giardia lamblia* leads to malabsorption due to duodenal villous atrophy. Histological abnormalities in CVID encompass intra-epithelial lymphocytosis, including influx of γδ IELs [78,79]. For this reason, it can be difficult to distinguish CeD from CVID. HLA genotyping can be used to exclude CeD because HLA-DQ2 is not overrepresented in CVID patients with villous atrophy. In addition, due to the impaired humoral immune response, patients with CVID are commonly unable to produce antibodies of any kind, including CD-related antibodies. The histological hallmark of CVID is the absence or sparsity of plasma cells in the lamina propria [80,81,82].

Eosinophilic gastroenteritis (EGE) is an uncommon disease characterized by eosinophilic infiltration of the gastrointestinal tract in the absence of secondary causes. Approximately 40% of patients with EGE have a history of an allergic disease, including asthma, defined food sensitivities, eczema, or rhinitis. Although most patients have positive skin test responses, typical anaphylactic reactions are absent, which suggests a delayed type of food hypersensitivity [83].

Clinical symptoms depend on the layer of the intestinal wall that is involved. When the mucosal and submucosal layer are predominantly involved, patients can develop malabsorption, protein-losing enteropathy, and failure to thrive. If the muscular propria layer is predominantly inflamed, this leads to bowel thickening, and patients can experience small bowel obstruction. Lastly, a serosal predominant form is recognized that is rare and can present with exudative ascites [83,84,85,86].

Graft-versus-host disease. Acute and chronic graft-versus-host disease (GVHD) occurs when immune cells transplanted from a nonidentical donor (the graft) recognize the transplant recipient (the host) as foreign, thereby initiating an immune reaction that causes disease in the transplant recipient [87]. Clinically significant acute GVHD occurs in patients who receive an allogeneic hematopoietic cell transplant (HCT) despite intensive prophylaxis with immunosuppressive agents. Gastrointestinal involvement usually presents with diarrhea and abdominal pain, but may also manifest as nausea, vomiting, and anorexia [88,89]. Damage to the mucosal barrier leads to translocation of microbial products and aggravates the inflammatory response. Confirmation of the diagnosis is provided by pathologic evaluation of tissue obtained by upper endoscopy, rectal biopsy, or colonoscopy.

Medication-related. Other drugs associated with the development of immune-mediated enteropathy have been reported. Among them, the following are worth mentioning:Ipilimumab. The drug is well known for gastrointestinal side effects including enterocolitis, which is probably initiated by T-cell activation in the gut [90].Immune checkpoint inhibitors such as anti-programmed death-1 antibodies (anti-PD-1, pembroluzimab, and nevolimab) are associated with a similar risk of immune mediated entero-(colitis) [91].Interferon α and interferon β 1, are drugs that contribute to CeD development in genetically susceptible patients [92,93].Mycophenolate mofetil is another drug that has been associated with new-onset diarrhea and villous atrophy in four patients who used this drug to prevent rejection after renal transplantation, with the complete recovery of symptoms after the cessation of the drug [94,95].Methotrexate, capecitabine, and TNFα inhibitors, indicating that clinicians and pathologists should be aware of the expanding spectrum of drugs that can cause apoptotic enteropathy [96].Olmesartan-associated enteropathy. Up to now, more than 100 cases of olmesartan-associated enteropathy have been described [78,79,80,81,82,83,84,85,86,87,88,89,90,91,92,93,94,95,96,97]. The mechanism by which olmesartan exerts its effect is not yet fully understood, but a role for cell-mediated immunity has been suggested based on the long delay between the onset of olmesartan therapy and the development of diarrhea (and enteropathy). Patients usually present with nonbloody diarrhea and weight loss, but some develop severe dehydration with electrolyte imbalances and acute renal failure. Up to 64% of patients carry the HLA-DQ2 and/or HLA-DQ8 gene, in comparison to only 40% in the general population; this suggests a genetic susceptibility in its pathogenesis [74]. Histopathology can mimic AIE, especially with respect to intra-epithelial lymphocytosis within or close to normal limits, loss of Paneth and goblet cells, and crypt apoptosis.


*Infectious*


Whipple’s disease is an exceedingly rare systemic infectious disease with an annual incidence of 3 in 1 million. This disease should be kept in mind in patients with abdominal pain, diarrhea, weight loss, and joint problems. The presence of vitamin or iron deficiency anemia, hypoalbuminemia, and relative lymphopenia should increase the level of suspicion. Other rare manifestations of Whipple’s disease include immunosuppressive treatment-resistant seronegative arthritis, unknown fever, chronic serositis, myoclonus, ophthalmoplegia, and early onset of cognitive disorders. Endocarditis caused by *T. whipplei* may not be associated with the classical clinical presentation of Whipple’s disease. In fact, affected patients may have no clinical or histologic evidence of gastrointestinal disease or arthralgias. Clinicians should be aware of this infectious disease in these conditions, as delay in diagnosis may be life-threatening [98,99,100]. The diagnosis of Whipple’s disease can be made with PAS-positive staining on a small bowel biopsy (Figure 3), or when two *T. whipplei* tests from gastrointestinal and/or extraintestinal specimens are positive.

Tropical sprue is a chronic diarrheal disease, possibly of infectious origin, that involves the small intestine and is characterized by the malabsorption of nutrients, especially folate and vitamin B12. The disease affects indigenous populations as well as visitors to the tropics who stay for more than a month; it is seldom seen in travelers who visit an endemic area for less than two weeks. Though no single identifiable pathogen has been identified as the cause of tropical sprue, it is likely that persistent overgrowth in the small intestine eventually causes significant small bowel structural damage and chronic diarrhea. Patients may have associated cramps, flatulence, and fatigue and, over time, progressive weight loss. Signs of malabsorption are common, including glossitis, cheilitis, distended abdomen, pallor, and pedal edema. tTGA levels are not elevated. The typically rapid clinical improvement with folate helps to establish the diagnosis. Complete histological improvement generally follows treatment with doxycycline, although this may take several months [101,102].

*Giardia lamblia* is the third most common agent of diarrheal disease in children under age five (after rotavirus and *Cryptosporidium* spp.); >300 million cases are reported annually, and it is a well-recognized cause of enteric disease among international travelers in the United States, Canada, and Europe. The pathogenic action of the protozoan and the inflammatory response activated in the host cause damage to the absorptive mucosa, with disruption of innate mucosal protective barriers. This results in malabsorption triggering symptoms such as diarrhea, abdominal pain, anorexia, and weight loss. As a result, patients will have many vitamin and mineral deficits such as vitamin A, thiamine, vitamin B12, folate, and iron, resulting in anemia. Acquired lactose intolerance occurs in up to 40 percent of patients; clinically, this manifests with exacerbation in intestinal symptoms following ingestion of dairy products. Recovery can take many weeks, even after clearance of the parasite [103,104].

Tuberculosis. Some cases of villous atrophy due to tuberculosis have been reported, and this cause should be considered in any patient from an endemic area presenting with diarrhea, fever, weight loss, and the presence of seronegative villous atrophy for celiac disease [105,106,107]. Increasing use of small bowel endoscopy finds ulcers in a relevant number of patients, mimicking the course of inflammatory bowel disease. Suspecting and treating this disease is of crucial importance to avoid a fatal outcome.

Human immunodeficiency virus. There is a growing body of evidence for small bowel pathology in patients infected with the human immunodeficiency virus (HIV) [108,109], and there is broad agreement that atrophy of villi is characteristic of such enteropathy. Furthermore, there is evidence that jejunal mucosal pathology has functional importance in reduced fat absorption. As a rule, HIV-infected patients with gastrointestinal symptoms show low-grade small bowel atrophy and a maturational defect in enterocytes, which may be caused exclusively by HIV. An additional intestinal infection can mask this mucosal atrophy [110,111].

Small intestinal bacterial overgrowth (SIBO) is a common gastrointestinal (GI) problem. Multiple predisposing factors have been recognized in the peer-reviewed literature, including achlorhydria, motility disorders, anatomical abnormalities of the gastrointestinal tract, and immunodeficiency disorders, including cancer. Commonly presenting complaints include abdominal distension, diarrhea, and malabsorption. The most consistent biopsy abnormality is villous blunting. Lamina propria inflammation and intraepithelial lymphocytosis are sometimes seen. The diagnosis can be confirmed by breath test analysis or, more definitively, by microbial analysis of jejunal aspirates [102,112]. Although symptom relief can be achieved through multiple-antibiotic regimens, the correction of underlying etiology, if possible, is necessary for long-lasting cure.

Viral/Postviral. There is direct evidence that celiac-like enteropathy can be initiated by norovirus infections and other viral infections, so it is likely that a transient viral enteritis could explain many of these cases. Neutrophil infiltration, epithelial degeneration, and crypt apoptosis are often found in biopsies in this scenario [113,114,115].

*H. pylori* rarely leads to celiac-like changes, usually causing lymphocytic enteritis (LE). LE represents a novel entity that is attracting increasing interest. *Helicobacter pylori* is one of the three most common causes of LE, along with the gluten-related disorders (celiac disease, nonceliac gluten sensitivity, and wheat allergy) and drug-related damage. The diagnosis of LE needs to be driven by predominant symptoms and patient history. However, it is often difficult to achieve an immediate identification of the underlying condition, and a broad variety of diagnostic tests may be required. Ultimately, long-term surveillance is needed for a final diagnosis in many cases, since a hidden or quiescent condition may be disclosed after a period of latency. In this scenario, strict collaboration between the clinician and the pathologist is pivotal [116].


*Infiltrative*


Collagenous sprue classically presents in middle-aged or elderly women, with persistent diarrhea, progressive weight loss, and severe malabsorption. Collagenous sprue is frequently associated with celiac disease, with a thick band of subepithelial collagen in the histopathology being characteristic in this case; however, their relationship remains controversial. Collagenous sprue may result in life-threatening complications such as ulcerative perforation and T-cell or B-cell lymphoma. Collagen deposits may also be found in gastric and/or colonic mucosa in patients with collagenous mucosal inflammatory diseases [117,118].

Amyloidosis. Gastrointestinal amyloidosis (GIA), a protein deposition disorder, represents a complex common pathway that encompasses multiple etiologies and presentations. Common presenting symptoms include weight loss, diarrhea, abdominal pain, malabsorption, esophageal reflux, and varying degrees of upper and lower GI bleeding, including fatal hemorrhage [119,120,121,122]. Hepatic symptoms include jaundice, steatorrhea, anorexia, and those related to portal hypertension such as ascites and splenomegaly. The gold standard for diagnosing amyloidosis is tissue biopsy of an affected organ with congo red stain demonstrating green birefringence under polarized light.


*Neoplastic*


Immunoproliferative small intestine lymphoma (IPSID), also known as Mediterranean lymphoma or α-heavy chain disease, is a rare extranodal marginal zone B-cell lymphoma occurring primarily in the proximal small intestine [123]. It is a variant of mucosa-associated lymphoid-tissue lymphoma (MALT) described in young adults from the developing world and is characterized by lymphoplasmocytic intestinal infiltrates with monotypic α-heavy chain expression. Its pathogenesis is attributed to antigenic stimulation by gastrointestinal infections, most notably *Campylobacter jejuni*, as identified by retrospective molecular analyses of tissue biopsies [124]. Typically, patients present with a malabsorption syndrome characterized by abdominal pain, weight loss, and biochemical abnormalities including hypoproteinemia and electrolyte derangements due to chronic diarrhea. Pathology shows extensive lymphoplasmacytic infiltration with blunting or absence of the villi and destruction of the crypts, and in later stages the infiltrate can be marked by atypical lymphoid cells [125].

Enteropathy-associated T-cell lymphoma (EATL) is a rare but severe complication of celiac disease (CeD), often preceded by low-grade clonal intraepithelial lymphoproliferation, referred to as type II refractory CeD (RCD-II) [126]. EATL can arise de novo in patients with CeD, but up to 50% of EATL develop through an intermediary RCDII step [127,128]. RCDII is characterized by the massive infiltration of the gut epithelium by lymphocytes, which display clonal rearrangements of the T-cell receptor (TCR) and, in most cases, an unusual immunophenotype combining both T-cell and NK-cell traits that reflects their origin from a small subset of innate like T-IEL [129]. EATL prognosis remains extremely poor, with a five-year survival rate of around 60% in de novo EATL, which decreases to less than 5% in patients with EATL complicating RCDII [130].

Diffuse large B-cell lymphoma is a dreaded complication of inflammatory bowel diseases (IBD). Knowledge about lymphoma in patients with IBD is limited to epidemiological data and the description of risk factors [131]. However, clinicians should bear in mind that Crohn’s disease may be complicated by neoplasia such as adenocarcinomas or lymphomas, especially if the disease duration is long and the patient is under immunosuppressive therapy [132]. The prognosis for lymphomas occurring in IBD appears to be good and what is expected, irrespective of the exposure to biologics and/or immunosuppressants [133].


*Diet-Related*


Severe protein malnutrition or kwashiorkor. Environmental enteric dysfunction, also known as malnutrition enteropathy [134], is associated with villous atrophy, mucosal thinning, increased intestinal permeability, loss of tight junction proteins (leading to loss of gut barrier function), lymphocytic infiltration, and gut dysbiosis [135,136,137]. Production of gastric acid is reduced, and exocrine pancreatic insufficiency is almost universal. The production of digestive enzymes and membrane nutrient transporters is also reduced. However, pancreatic enzyme replacement therapy does not improve weight gain. Lactase deficiency is common, leading to lactose malabsorption. Luminal bacterial overgrowth is common, and the diminished gut barrier function, which normally limits translocation of bacteria and their toxins, can lead to bacteremia and sepsis [135,136,137].


*Ischemic*


Chronic mesenteric ischemia. Villous atrophy can occur in some cases of chronic mesenteric ischemia (CMI). Progressive atherosclerotic disease occurs in more than 90% of cases of chronic mesenteric ischemia [138]. Patients with chronic CMI present with typical symptoms such as postprandial abdominal pain (typically 30–60 min after eating), early satiety, diarrhea, or constipation (or both), nausea or vomiting (or both), a fear of eating, and subsequent weight loss. While postprandial pain is associated with other abdominal diseases, including gastric reflux, peptic ulcer disease, biliary disease, pancreatitis, diverticular disease, inflammatory bowel disease, and gastroparesis, the predominant symptom for chronic CMI is weight loss [139].


*Others*


Radiation enteritis. Radiation therapy (RT) can cause acute injury to the small and large intestines that develops during or shortly after treatment of a variety of malignancies. The initial toxicity generally resolves within a matter of weeks, but chronic changes (mucosal atrophy as well as collagen deposition and fibrosis) can develop months or years after therapy [140]. These chronic changes can impair the absorption of fats, carbohydrates, protein, bile salts, and vitamin B12, leading to loss of water, electrolytes, and proteins in the small intestine. Bile salt resorption may be impaired, causing increased amounts of conjugated bile salts to enter the colon. Intestinal failure requiring either surgery or home parenteral nutrition develops in approximately 5% of patients treated with radiation [141].

### 2.4. Postabsorptive, Processing Phase

ApoB48 is used by enterocytes in the assembly of chylomicrons. ApoB48 serves as an acceptor for newly synthesized triglycerides being transferred by MTP, which is necessary to transfer triglycerides formed in the endoplasmic reticulum. Mutations in the gene for MTP are the basis for A-β-lipoproteinemia, which is characterized by the absence of intestinal lipoproteins in plasma [2].

#### 2.4.1. Enterocyte Processing

##### Abetalipoproteinemia

Lipoprotein particles are mainly assembled in enterocytes and hepatocytes. Enterocytes assemble and secrete chylomicrons to transport dietary fat and fat-soluble vitamins. Hepatocytes produce very low density lipoproteins (VLDL). Assembly of these particles is dependent on two proteins:143 apoB, a structural protein, and microsomal triglyceride transfer protein (MTP), which helps to assemble apoB-containing lipoproteins. Loss-of-function mutations in the MTTP gene cause FHBL-SD1, commonly known as abetalipoproteinemia [142,143,144,145]. Mutations in the APOB gene disable apoB protein from forming a lipoprotein and result in FHBL-SD2, with homozygous and heterozygous forms [146].

Hallmark symptoms are fat malabsorption or steatorrhea, spinocerebellar degeneration, acanthocyte red blood cells, and retinitis pigmentosa. These patients may also show symptoms of deficiencies of lipid-soluble vitamins (A, E, D, and K) [147,148].

#### 2.4.2. Lymphatic

##### Intestinal Lymphangiectasia

Primary intestinal lymphangiectasia (PIL) is a rare disorder of unknown etiology resulting in protein-losing enteropathy [149,150]. Although usually developing in early childhood, cases have also been described in adults [149]. The disease is caused by a diffuse or localized dilatation and/or rupture of intestinal lymphatic vessels in the mucosa, submucosa, or subserosa due to high pressure in lymphatic vessels. The dilatation and leakage of intestinal lymph vessels lead to hypoalbuminemia, hypogammaglobulinemia, and lymphopenia. The range of associated symptoms is wide, from mild lower-limb edema to generalized edema, abdominal (chilous) and/or pleural or epicardial effusion, and recurrent diarrhea [151,152]. Pathologies causing secondary intestinal lymphangiectasia include tuberculosis, sarcoidosis, eosinophilic gastroenteritis, systemic sclerosis, systemic lupus erythematosus retroperitoneal fibrosis, and constrictive pericarditis.

##### Radiation

Lymphatic ectasia as a histological feature has been described in association with postradiotherapy malabsorption [153,154,155]. This observation underscores the importance of the endoscopic observation of the duodenum after irradiation to the abdomen because such treatment may cause intestinal lymphangiectasia and subsequent protein-losing enteropathy [153,156]. Some of these cases are asymptomatic.

## 3. Consequences of Nutrient Malabsorption

The clinical and nutritional consequences of malabsorptive syndromes depend on the condition causing malabsorption and the phase of absorption that is impaired (Table 1, Table 2 and Table 3).

In some cases, poor assimilation of nutrients depends on maldigestion, which refers to impaired digestion of nutrients within the intestinal lumen (i.e., exocrine pancreatic insufficiency; failure in the concentration of bile salts in the duodenum‒jejunal lumen) or at the terminal digestive site of the brush border membrane of mucosal epithelial cells (i.e., lactose maldigestion). In other cases, malabsorption depends exclusively on the presence of a disorder that alters the morphology of the intestinal mucosa, reducing its ability to transport nutrients through the cell interior (i.e., celiac disease) or its incorporation into the general circulation through lymphatic vessels (i.e., lymphangiectasia).

Finally, malabsorption may be either global or selective. In the first case, malabsorption arises from diseases associated with either widespread mucosal involvement (i.e., CeD, amyloidosis, mastocytosis) or a reduced absorptive surface (i.e., extensive bowel resection). In such cases, malabsorption can affect a wide variety of macronutrients (fats, carbohydrates, and proteins) and micronutrients (vitamins, minerals, and trace elements).

In contrast, some diseases interfere with the absorption of a single nutrient or a limited array of nutrients (i.e., vitamin B12 deficiency in pernicious anemia or resection of more than 30 cm of the terminal ileum).

Advanced-stage chronic pancreatitis (CP) represents the paradigm of disease that causes maldigestion, while celiac disease is the most common example of a malabsorptive condition. Both diseases explain the nutritional deficiencies most often observed in the gastroenterologist’s office (Table 1, Table 2 and Table 3)

Patients with pancreatic exocrine insufficiency (PEI) may have maldigestion-related symptoms (i.e., diarrhea, flatulence, abdominal distension, cramps, and weight loss). Patients tend to adapt their dietary habits to avoid or minimize symptoms, and that explains why patients with PEI are often asymptomatic [157]. Weight loss should be considered over body mass index (BMI) as a stand-alone measurement as, although BMI changes over time, the widespread presence of obesity renders a single measurement of BMI clinically irrelevant. In fact, sarcopenia can occur in the presence of obesity and has a prognostic role in pancreatic diseases [158,159]. The degree of malnutrition in CP ranges from overnutrition (obesity) to severe malnutrition.

Low circulating levels of fat-soluble vitamins (A, D, E, and K), proteins (albumin, prealbumin, retinol binding protein, and transferrin), lipoproteins and apolipoproteins, and mineral trace elements (magnesium, zinc, calcium, iron, and selenium) have been reported in patients with CP [160,161,162,163,164]. Most of these abnormalities are related to PEI. However, there are other factors that contribute to malnutrition in chronic pancreatitis, such as alcohol abuse, symptoms limiting food ingestion (abdominal pain), and complications including alterations of the gastroduodenal transit secondary to compression from pseudocysts or duodenal fibro-inflammatory involvement, as well as cholestasis due to compression or infiltration of the common bile duct (Figure 1). Water-soluble vitamins are rarely deficient in CP, except for cobalamin, which is reduced in the presence of severe insufficiency [165,166]. Section 4 describes in more detail the nutritional consequences observed in chronic pancreatitis.

CeD affects 1% of the population, but the diagnosis is often missed or delayed. Although diarrhea remains a common presenting complaint, it is increasingly recognized that CeD can present without gastrointestinal symptoms, and often the symptoms and/or signs depend on the nutrient deficiency. Such is the case with anemia, osteoporosis, and various forms of neuropathy.

The major site of damage is in the proximal small intestine. This damage causes a wide variety of consequences, including maldigestion and malabsorption. These derangements result in the characteristic, although not universal, features of malnutrition [167]. Again, the degree of malnutrition in CeD ranges from severe florid malabsorption cases with wasting to unsuspected forms of malabsorption of a few nutrients (Table 1, Table 2 and Table 3).

**Table 1 nutrients-13-01254-t001:** Causes and nutritional consequences of malabsorption (fat-soluble vitamins) [168,169,170,171,172,173,174,175,176,177].

Specific Nutritional Deficiency	Causes	Symptoms and Signs Due to Micronutrient Maldigestion or Malabsorption
**Vitamin A ^1^**	Bariatric surgery (gastric bypass, biliopancreatic, or duodenal switch procedures) Disorders associated with fat malabsorption: Cystic fibrosis and other causes of pancreatic insufficiency Celiac disease Cholestatic liver disease such as: ◦ Primary biliary cholangitis ◦ Primary sclerosing cholangitis ◦ Familial intrahepatic cholestasis Small bowel Crohn’s disease Short bowel syndrome Congenital intestinal lymphangiectasia Infiltrative diseases such as amyloid or lymphoma Abetalipoproteinemia	**[A]*****Serum retinol levels (levels less than 20 μg/dL (0.7 μmol/L) suggest deficiency.*** Xerophthalmia Night blindness (nyctalopia) and retinopathy Poor bone growth Hyperkeratosis, phrynoderma (follicular hyperkeratosis) Impairment of the humoral and cell-mediated immune system via direct and indirect effects on the phagocytes and T cells
**Vitamin D ^2,3,4^**		**[D] *Deficiency (serum 25[OH]D <12 ng/mL (30 nmol/L) or insufficiency (12 to 20 ng/mL (30 to 50 nmol/L)*** Reduced intestinal absorption of calcium and phosphorus Secondary hyperparathyroidism and phosphaturia Bone pain and tenderness, muscle weakness Low bone mass on bone densitometry and fractures
**Vitamin K ^5^**		**[K]*****Vitamin K status also can be determined indirectly by measuring vitamin K-dependent factors (i.e., prothrombin, factors VII, IX, X, or protein C)*** Easy bruisability Mucosal bleeding Splinter hemorrhages Epistaxis, melena, hematuria, and any other manifestations of impaired coagulation
**Vitamin E ^6^**		**[E]*****Defined as below 0.5 mg/dL*** Neuropathy (spinocerebellar syndrome) Skeletal myopathy Pigmented retinopathy Brown bowel syndrome (intestinal lipofuscinosis) Hemolytic anemia

^1^ Serum retinol levels (levels less than 20 μg/dL (0.7 μmol/L) suggest deficiency. Serum retinol levels may be artificially low (i.e., underestimated vitamin A stores) in severe systemic inflammatory disease and severe malnutrition. ^2^ Other causes of Vitamin D deficiency include decreased intake, reduced sun exposure, increased hepatic catabolism, decreased endogenous synthesis (via decreased 25-hydroxylation in the liver or 1-hydroxylation in the kidney), or end-organ resistance to vitamin D. ^3^ In addition to its role in calcium and bone homeostasis, vitamin D could potentially regulate many other cellular functions, but a causal association between poor vitamin D status and nearly all major diseases (cancer, infections, autoimmune diseases, and cardiovascular and metabolic diseases) has not been established. ^4^ Patients with vitamin D deficiency (serum 25[OH]D <12 ng/mL (30 nmol/L)) or insufficiency (12 to 20 ng/mL (30 to 50 nmol/L)) should receive vitamin D supplementation to treat the deficiency/insufficiency. ^5^ Other causes of Vitamin D deficiency include liver failure, second- and third-generation cephalosporin antibiotics, remarkably high doses of vitamin E, and toxic doses of vitamin A. ^6^ Vitamin E deficiency is uncommon, except in patients with severe and prolonged cholestasis.

**Table 2 nutrients-13-01254-t002:** Causes and nutritional consequences of malabsorption (water-soluble vitamins) [168,169,170,171,172,173,174,175,176,177].

Specific Nutritional Deficiency	Causes	Symptoms and Signs Due to Micronutrient Maldigestion-Malabsorption.
**Vitamin B1** (thiamine)	Anorexia nervosa Bariatric surgery Malabsorptive syndromes such as active celiac disease, malignancies, and short bowel syndrome	**[B1] *Measurement:****the normal range for blood thiamine concentration varies somewhat between laboratories, 70 to 180 nmol/L (3.0 to 7.7 mcg/dL).* Thiamine deficiency in the diet causes two clinical phenotypes: Beriberi (infantile and adult) Wernicke‒Korsakoff syndrome
**Vitamin B2** (Riboflavin) *Ariboflavinosis*		**[B2] *Measurement:****E**rythrocyte glutathione reductase assay is a better functional index of insufficient riboflavin intake (a coefficient >1.4 indicates riboflavin insufficiency).* Symptoms of ariboflavonosis: Sore throat, hyperemia of pharyngeal mucous membranes, edema of mucous membranes, cheilitis, stomatitis, glossitis, normocytic-normochromic anemia, and seborrheic dermatitis
**Vitamin B3** (Niacin) ^1^		**[B3] *Measurement:****High levels of a metabolic product of a vitamin, such as N-methylnicotinamide, reflect adequate concentrations of intracellular niacin.* Photosensitive pigmented dermatitis (typically located in sun-exposed areas), diarrhea, and dementia
**Vitamin B5** (Pantothenic acid)		**[B5] *Measurement:****Urinary excretion below 1 mg/day generally indicates low dietary intake* Paresthesias and dysesthesias, referred to as “burning feet syndrome”
**Vitamin B6** (Pyridoxine) ^2^		**[B6] *Measurement:****20 to 30 nmol/L (4.9 to 7.4 ng/mL) is generally considered marginal and >30 nmol/L (>7.4 ng/mL) is sufficient.* Marginal deficiencies: nonspecific stomatitis, glossitis, cheilosis, irritability, confusion, depression, and possibly peripheral neuropathy. Severe deficiency is associated with seborrheic dermatitis, microcytic anemia, and seizures.
**Biotin**		**[Biotin] *Measurement:****Normal urine biotin excretion is around 75 to 195 μmol/L day* Dermatitis around the eyes, nose and mouth, conjunctivitis, alopecia, and neurologic symptoms (changes in mental status, lethargy, hallucinations, and paresthesias).
**Vitamin C (Ascorbic acid)** ^3^		**[Vitamin C] *Measurement:****Symptoms of scurvy generally occur when the plasma concentration of ascorbic acid is less than 0.2 mg/dL (11 μmol L)* Follicular hyperkeratosis and perifollicular hemorrhage, with petechiae and coiled hairs. ecchymoses, gingivitis (with bleeding and receding gums and dental caries). Sjogren’s syndrome, arthralgias, edema, anemia, and impaired wound healing. Musculoskeletal pain may be caused by hemorrhage into the muscles or periosteum. Generalized systemic symptoms are weakness, malaise, joint swelling, arthralgias, anorexia, depression, neuropathy, and vasomotor instability.
**Vitamin B12 and folate** ^4^	Decreased intake (e.g., reduced intake of animal products, strict vegan diet, breastfeeding by a vitamin B12-deficient mother) Autoantibodies to intrinsic factor or gastric parietal cells (e.g., pernicious anemia) Decreased absorption (e.g., gastrectomy, bariatric surgery, pancreatic insufficiency, celiac disease, Crohn’s disease, bacterial overgrowth, fish tapeworm infection, gastric atrophy associated with aging, extensive ileal resection, or bypass) Medications and drugs that interfere with absorption or stability (e.g., metformin, histamine receptor antagonists, proton pump inhibitors, nitrous oxide)	**[Cobalamin] *Measurement:****Below 200 pg/mL (below 148 pmol/L)—low; consistent with deficiency; 200 to 300 pg/mL (148 to 221 pmol/L)–borderline; deficiency is possible and additional testing is useful.* **[Folate] *Measurement:****From 2 to 4 ng/mL (from 4.5 to 9.1 nmol/L)—borderline: Below 2 ng/mL (below 4.5 nmol/L)—low, consistent with folate deficiency* Macrocytic anemia, mild leukopenia and/or thrombocytopenia, low reticulocyte count, hypersegmented neutrophils Cognitive slowing and neuropathy Fatigue, irritability Cognitive decline, forgetfulness, dementia, psychosis Glossitis, oral ulcers (folate deficiency) Symmetric paresthesias or numbness and gait problems Restless legs syndrome Subacute combined degeneration of the dorsal (posterior) and lateral columns (white matter) of the spinal cord due to demyelination (weakness, ataxia, and paresthesias that may progress to spasticity and paraplegia) Abnormal deep tendon reflexes Extrapyramidal signs (i.e., dystonia, dysarthria, rigidity) Skin hyperpigmentation Increased risk of osteoporosis and fractures of the hip or spine Increased risk of gastric cancer in individuals with pernicious anemia (B12 deficiency) Risk of neural tube defects (folate deficiency)

^1^ Other causes of vitamin B3 (niacin) deficiency include carcinoid syndrome, prolonged use of isoniazid, and Hartnup disease. ^2^ Decreased concentrations of pyridoxal-5-phosphate have been also reported in asthma, diabetes, alcoholism, heart disease, pregnancy, breast cancer, Hodgkin’s lymphoma, and sickle-cell anemia. ^3^ Ascorbic acid deficiency occurs mostly in severely malnourished individuals, drug, and alcohol abusers, or those living in poverty or on diets devoid of fruits and vegetables. ^4^ A normal MCV does not exclude vitamin B12 or folate deficiency.

Several factors contribute to the diarrhea associated with CeD [178]:Secretin and cholecystokinin release in response to a meal are impaired in CeD, thus diminishing the delivery of bile and pancreatic secretion into the duodenal lumen.The delivery of excessive dietary fat into the colon results in bacterial production of hydroxy fatty acids, which are potent cathartics.The stool volume and osmotic load delivered to the colon are increased by malabsorption.There is active electrolyte secretion in the lumen of the small intestine.If the disease extends to and involves the ileum, patients can experience the direct cathartic action of malabsorbed bile salts on the colon.

The weight loss typically seen in the celiac patient may be masked by a noticeable and unexpected increase in appetite. In some debilitated patients, some weight loss may be masked by fluid retention caused by hypoproteinemia.

Malaise, lassitude, and fatigue are also common even when anemia is absent.

Exceptionally, CeD may manifest quite abruptly with profuse diarrhea and severe malabsorption syndrome and related electrolyte abnormalities, hypoproteinemia, and, in some instances, hemodynamic instability (a “celiac crisis”). This would imply that even a typically chronic disorder, such as CeD, may have acute onset in a small proportion of patients, which emergency physicians should be aware of. Although rarely encountered in clinical practice, this acute onset of CeD requires hospitalization and immediate treatment (i.e., electrolyte replacement and protein correction) to avoid life-threatening complications [179].

Additional features associated with celiac disease are low birth weight and intrauterine growth retardation among infants born to untreated celiac mothers, recurrent abortions, infertility, late menarche, early menopause, persistent transaminitis, short stature, premature osteopenia, and osteoporosis, aphthous stomatitis, folate-zinc deficiency, macrocytosis, depression, and chronic fatigue [167,180].

Initial evaluation for newly diagnosed celiac disease includes testing for iron, folate, and vitamin B12 deficiencies.

### 3.1. Iron

Iron is an essential micronutrient; it is required for adequate erythropoietic function, oxidative metabolism, enzymatic activities, and cellular immune responses.

Iron deficiency (ID) and iron deficiency anemia (IDA) in CeD can occur in the absence of other malabsorptive manifestations and might be the presenting feature. The frequency of iron deficiency anemia in CeD varies from 12% to 69% [181,182,183]. One in 31 patients with IDA could have CeD [181], with this risk unaffected by sex, age, or prevalence of celiac disease in the underlying population, indicating that CeD is a consistent risk factor for IDA.

Malabsorption is likely the cause of iron deficiency anemia, and results in fatigue and diminished muscular oxygenation, which may affect muscle strength and quality and, subsequently, physical performance. Other causes may contribute, such as occult gastrointestinal bleeding and anemia of chronic process (ACP) [184,185,186]. ACP is an old concept in the scientific literature, but current research on the role of pro-inflammatory cytokines and iron metabolism has yielded more information about the pathophysiology of this disease. This type of anemia responds to a multifactorial pathogenesis including four fundamental mechanisms: abnormalities in iron utilization, a decrease in the half-life of red blood cells, direct inhibition of hematopoiesis, and relative deficiency of erythropoietin [181].

Iron deficiency anemia resolves with adherence to a gluten-free diet, although normalization of the iron stores may require months, in tandem with the healing of the small intestinal mucosa. Oral iron is the first line of treatment in uncomplicated ID, but the threshold for use of parenteral iron in cases of moderate or severe anemia, severe clinical symptoms, lack of response to oral iron replacement (common in patients with villous atrophy), intolerable adverse effects, or difficult adherence is lowering.

### 3.2. Folate and Cobalamin

Folate (vitamin B9) and cobalamin are both water-soluble B vitamins required for the formation of hematopoietic cells (red blood cells, white blood cells, and platelets).

Usually, people suffering from CeD can develop folate and vitamin B12 deficiencies because of generalized malabsorption linked to villi atrophy. Both vitamins are essential for normal hematopoiesis and neurologic function (Table 2).

Deficiency of vitamin B12 is common in CeD and frequently results in anemia. Though the terminal ileum is the primary site of absorption of vitamin B12, García-Manzanares and Lucendo [187] reported a prevalence of vitamin B12 deficiency of between 8% and 41% in patients with newly diagnosed CeD. The causes of B12 deficiency in CD are still not clear, but they may be related to complications of small intestinal injury including decreased gastric acidity, cobalamin intake due to the frequent finding of bacterial overgrowth, autoimmune gastritis, and decreased efficiency of the intrinsic factor or even dysfunction of the distal small intestine [188]. A recent study of histologic changes in CeD by Dickey and Hughes revealed an increase in the degree of ileal intraepithelial lymphocytosis, and this finding was correlated to duodenal villous atrophy [189]. This underlying mucosal damage could help explain the origin of vitamin B12 deficiency in celiac disease [190].

Folate absorption occurs primarily in the jejunum, which is commonly affected by CeD. Prior to uptake, folate must be deconjugated by a brush border membrane peptidase, and the intestinal mucosa damage in CeD may affect enzyme activity, leading to folate deficiency. Several studies in adult celiac patients have shown an increased risk of folate deficiency, which can affect up to 20–30% of newly diagnosed patients [187,191]. Given these results, folate supplementation is recommended to ensure the minimum recommended requirements are consumed until the damaged, functionally impaired villous architecture normalizes in the absence of gluten. Both folate and vitamin B12 deficiencies can lead to a macrocytic anemia with low values for hemoglobin or hematocrit, and high mean corpuscular volume levels. However, MCV can be normal when iron deficiency coexists.

The classic findings associated with vitamin B12 and folate deficiency include worsening macrocytic anemia, yellow skin, and variable neurologic abnormalities more prominent in vitamin B12 deficiency (cognitive slowing and neuropathy). Neuropsychiatric symptoms can be present even in the absence of anemia or macrocytosis, and the lack of these hematologic changes cannot be used to exclude vitamin B12 deficiency as a cause of neuropsychiatric symptoms [192]. A typical symptom of cobalamin deficiency is glossitis (including pain, swelling, tenderness, and loss of papillae of the tongue), while folate deficiency can cause oral ulcers (Table 2).

### 3.3. Fat-Soluble Vitamins

In patients presenting with the classic malabsorption of celiac disease, deficiencies in fat-soluble vitamins (vitamins A, D, E, and K) are commonly encountered (Table 1).

Vitamin A is a subclass of a family of lipid-soluble compounds referred to as retinoic acids. In the eye, vitamin A has two major roles: prevention of xerophthalmia (abnormalities in corneal and conjunctival development) and phototransduction. Vitamin A is also crucial to cellular differentiation and integrity in the eye. Vitamin A deficiency’s clinical manifestations include night blindness, conjunctival dryness, and keratomalacia. Vitamin A deficiency is also associated with poor bone growth, nonspecific dermatologic problems (i.e., hyperkeratosis), and impaired immune function [168,193].

Vitamin D is ingested in a prohormone form, converted in the liver to 25-hydroxyvitamin D, and then later changed to the active form 1,25-hydroxyvitamin D at the level of the kidneys [194]. The recommended range of 25-hydroxyvitamin D is 30 to 50 ng/dL. The clinical manifestations of vitamin D deficiency depend on the severity and duration of the deficiency. Most patients with serum 25(OH)D between 12 and 20 ng/mL (37.5 and 50 nmol/L) are asymptomatic. With prolonged severe vitamin D deficiency, there is reduced intestinal absorption of calcium and phosphorus, and hypocalcemia occurs, causing secondary hyperparathyroidism, which leads to phosphaturia, demineralization of bones, and, when prolonged, osteomalacia in adults, which clinically manifests as muscle weakness and musculoskeletal pain [195]. Measurement of bone density, serum calcium, alkaline phosphatase, and parathyroid hormone levels is thereby recommended at the time of diagnosis. Additional recommended measures include ensuring adequate calcium intake to maintain 1500 mg per day, adhering to a strict gluten-free diet, exercise, abstaining from tobacco usage, minimizing alcohol intake, obtaining a baseline bone mineral density by dual energy X-ray absorptiometry scan, and screening for vitamin D deficiency. The risk of fractures has been shown to be increased in celiac patients (Figure 2) [196,197,198,199].

Vitamin E (as alpha tocopherol) works as a free radical scavenger, protecting polyunsaturated fatty acids, a major structural component of the cell membranes, from peroxidation [200]. Some functions of vitamin E are independent of the antioxidant/radical scavenging activity, including inhibition of cell proliferation, platelet aggregation, and monocyte adhesion [201]. Like other fat-soluble vitamins, the bioavailability of alpha-tocopherol depends on the physiological mechanisms of fat digestion and absorption. This process requires lingual and gastric lipases; bile salts for solubilization and production of mixed micelles; pancreatic function [202] and intestinal and ileal mucosal and absorptive mechanisms. In patients with CeD, malabsorption of vitamin E can occur due to both mucosal involvement and EIP. Vitamin D deficiency leads to hemolytic anemia, gait disturbance, and neuropathy (Table 1) [167].

Vitamin K has a major role in coagulation pathways because it is a cofactor required for the activity of several key proteins containing carboxyglutamic acid residues. Several other proteins within the body (i.e., osteocalcin) also contain carboxyglutamate residues and depend on vitamin K for their activity [168]. Clinical signs and symptoms of vitamin K deficiency include easy bruisability, mucosal bleeding, splinter hemorrhages, melena, hematuria, or any other manifestations of impaired coagulation. Low vitamin K levels are detected by prolonged prothrombin time and result in decreased synthesis of clotting factors II, VII, IX, and X.

### 3.4. Other Micronutrient Deficiencies

In addition to the above-mentioned deficiencies (i.e., iron, folic acid, and vitamin B12), at the time of diagnosis of CeD there may be deficiencies of other vitamins and minerals, in particular copper, zinc, and selenium (Table 3).

Copper is absorbed in the proximal small intestine and stomach. Absorption occurs by a saturable active transport process at lower levels of dietary copper and by passive diffusion at high levels of dietary copper. Absorbed copper is loosely bound to plasma albumin and amino acids in the portal blood and taken to the liver, where most of it is taken up on the first pass. In the liver, the copper is incorporated into the copper-containing protein ceruloplasmin, which serves to transport copper from the liver to peripheral tissues [203,204,205]. Copper is used as a cofactor by multiple enzymes involved in redox reactions and is essential for ceruloplasmin ferroxidase function. These cupro-enzymes help to explain some of the clinical features of severe copper deficiency, including a lack of skin pigmentation (decreased dopamine beta-hydrolase), weakness (decreased cytochrome c), and bleeding disorders (decreased factor V).

The exact prevalence of copper deficiency in active CeD is unclear, but conditions associated with malabsorption of macronutrients and gastrointestinal disease can impair copper uptake and contribute to suboptimal copper status. This deficiency can lead to anemia, thrombocytopenia, neutropenia, peripheral neuronal involvement, and myelopathy [206,207,208,209,210,211].

**Table 3 nutrients-13-01254-t003:** Causes and nutritional consequences of malabsorption (main minerals and trace elements) [207,208,209,210,211,212,213,214,215].

Specific Nutritional Deficiency	Causes	Symptoms and Signs Due to Micronutrient Maldigestion-Malabsorption
**Iron deficiency ^1^**	Disorders that affect the mucosal cells responsible for iron absorption, such as celiac disease, atrophic gastritis, *Helicobacter pylori* infection, and bariatric surgery, proton pump inhibitors^1^	**[Iron] *Measurement:****Transferrin saturation <19%; Ferritin 30 ng/mL (a higher ferritin level may be “falsely normal” and cannot be used to eliminate the possibility of iron deficiency in individuals with comorbidities).* Fatigue, Pica (Pagophagia), restless legs syndrome, headache, exercise intolerance, exertional dyspnea, weakness. Pallor, dry or rough skin, atrophic glossitis with loss of tongue papillae, cheilosis (also called angular cheilitis), koilonychia (spoon nails), esophageal web, which may be accompanied by dysphagia (e.g., Plummer‒Vinson or Patterson‒Kelly syndrome; rare), alopecia (rare) in especially severe cases, chlorosis (pale, faintly green complexion; extremely rare).
**Calcium deficiency**	(See causes of steatorrhea and vitamin D malabsorption) Severe acute pancreatitis	**[Calcium] *Measurement:****The normal range for serum calcium is approximately 8.6 to 10.2 mg/dL (2.15 to 2.54 mmol/L).* Neuromuscular irritability (paresthesias), carpopedal spasm, tetany, seizures, Heart failure or prolonged QT interval.
**Magnesium**	Acute or chronic diarrhea, malabsorption and steatorrhea, and small bowel bypass surgery Severe acute pancreatitis Proton pump inhibitors (in patients taking diuretics). Calcineurin inhibitors Alcohol Uncontrolled diabetes mellitus Posttransplant patients Hypercalcemia	**[Magnesium] *Measurement:****Hypomagnesemia is defined as a serum magnesium less than 1.6 mg/dL [0.66 mmol/L])* Neuromuscular manifestations, including neuromuscular hyperexcitability (e.g., tremor, tetany, convulsions), weakness, apathy, delirium, and coma Insulin resistance and the metabolic syndrome Migraine headaches and asthma Cardiovascular manifestations, including widening of the QRS and peaking of T waves with moderate magnesium depletion, and widening of the PR interval, diminution of T waves, and atrial and ventricular arrhythmias with severe depletion Abnormalities of calcium metabolism, including hypocalcemia, hypoparathyroidism, parathyroid hormone (PTH) resistance, and decreased synthesis of calcitriol Hypokalemia
**Copper**	Foregut surgery, including gastrectomy or gastric bypass Chronic diarrhea or other malabsorptive conditions including celiac disease Chronic peritoneal dialysis or hemodialysis Excessive zinc ingestion (copper and zinc are competitively absorbed from the jejunum) Total parenteral nutrition (TPN) with insufficient copper	**[Copper] *Measurement:****The normal range for total copper in the blood is 85 to 180 μcg/dL.* Fragile, abnormally formed hair, depigmentation of the skin, muscle weakness (myeloneuropathy), neurologic abnormalities, edema, and hepatosplenomegaly, and osteoporosis. The neurologic manifestations (ataxia, neuropathy, and cognitive deficits) can mimic vitamin B12 deficiency. Hematologic features: anemia (usually normocytic; sometimes macrocytic and occasionally with microcytic cells) and neutropenia. Thrombocytopenia may also occur but is relatively rare.
**Zinc ^2^**	Gastric bypass for obesity Necrotizing enterocolitis Malabsorption syndromes (such as celiac disease and chronic inflammatory bowel disease) Alcohol-related cirrhosis Medications that increase urinary losses of zinc, including thiazides, loop diuretics, and angiotensin receptor blockers Patients receiving chronic TPN solutions lacking adequate zinc supplementation.	**[Zinc] *Measurement:****The normal range for zinc in the blood is 64.7–139.8 μcg/dL* *Alkaline phosphatase activity can serve as a supportive marker of zinc status.* Delayed sexual maturation, impotence, hypogonadism, oligospermia Alopecia, dysgeusia (impaired taste) Immune dysfunction Night blindness Impaired wound healing Skin lesions (erythematous, vesiculobullous, and pustular lesions) Change in hair color and hair loss Impaired appetite Decubitus ulcers
**Selenium**	Gastric bypass for obesity Chronic diarrhea or other malabsorptive conditions including celiac disease Extensive intestinal resection TPN with insufficient copper	**[Selenium] *Measurement:****The normal range for selenium in the blood is 70**‒**150 ng/mL (0.15 parts per million)* Skeletal muscle dysfunction Cardiomyopathy Mood disorders Impaired immune function Macrocytosis Whitened nailbeds

^1^ If an individual with iron deficiency is taking a proton pump inhibitors, antacid, or histamine receptor blocker, we do not attribute iron deficiency to the medication without performing an evaluation for bleeding or reduced iron absorption, as indicated for the individual. ^2^ This disorder is caused by variants in the *SLC39A4* gene, which encodes a protein that appears to be involved in zinc transport and is characterized by signs and symptoms of severe zinc deficiency, including diarrhea, dermatitis (especially perioral and perianal), alopecia, poor growth, and poor immune function.

Zinc is absorbed mainly in the duodenum and jejunum, and to a lesser extent in the ileum and large intestine [212,213]. During digestion, dietary zinc is released and forms complexes with different ligands, namely amino acids, phosphates, organic acids, and histidines [214]. Zinc‒ligand complexes are then absorbed through the intestinal mucosa by both an active and passive process. Once absorbed, the portal circulation carries zinc to the liver [215]. Zinc absorption may be impaired in exocrine pancreatic insufficiency and in conditions associated with malabsorption. Fractional zinc is used as a cofactor for multiple metalloenzymes, and mild deficiency states cause impaired immune function, hypogonadism, oligospermia, alopecia, poor wound healing, and skin changes (i.e., pustular and vesiculobullous lesions) [167]. Nevertheless, the presence of clinical alterations because of zinc deficiency is uncommon in CeD.

Selenium is an ultra-trace element with a role in multiple biological functions. Selenomethionine is actively absorbed in the small intestine by the methionine absorptive pathway, and inorganic selenium (in supplements) is passively absorbed in the duodenum. Selenium absorption may be impaired in malabsorptive syndromes such as active celiac disease and short bowel syndrome.

CeD patients may be deficient in selenium, and there is evidence that products based on less popular grains, especially on oat, amaranth, teff, and quinoa, should be more frequently chosen as a source of selenium by people on a gluten-free diet than traditionally consumed gluten-free grains such as corn, rice, and buckwheat [167].

## 4. Conclusions

The normal uptake of nutrients, vitamins, and minerals by the gastrointestinal tract (GI) requires several steps, each of which can be compromised in disease. In healthy conditions, the GI tract will use nutrients, provide energy, and release wastes. However, numerous disorders can alter the physiological mechanisms that guarantee proper digestion and absorption of nutrients (macro-and micro-nutrients), leading to a wide variety of symptoms and nutritional consequences. Knowing the mechanisms that lead to poor assimilation of nutrients, as well as the symptoms and nutritional consequences of each specific disorder, adds value to the role of a wide group of professionals who participate in its management, including RDN, family doctors, internists, gastroenterologists, and surgeons.

## Figures and Tables

**Figure 1 nutrients-13-01254-f001:**
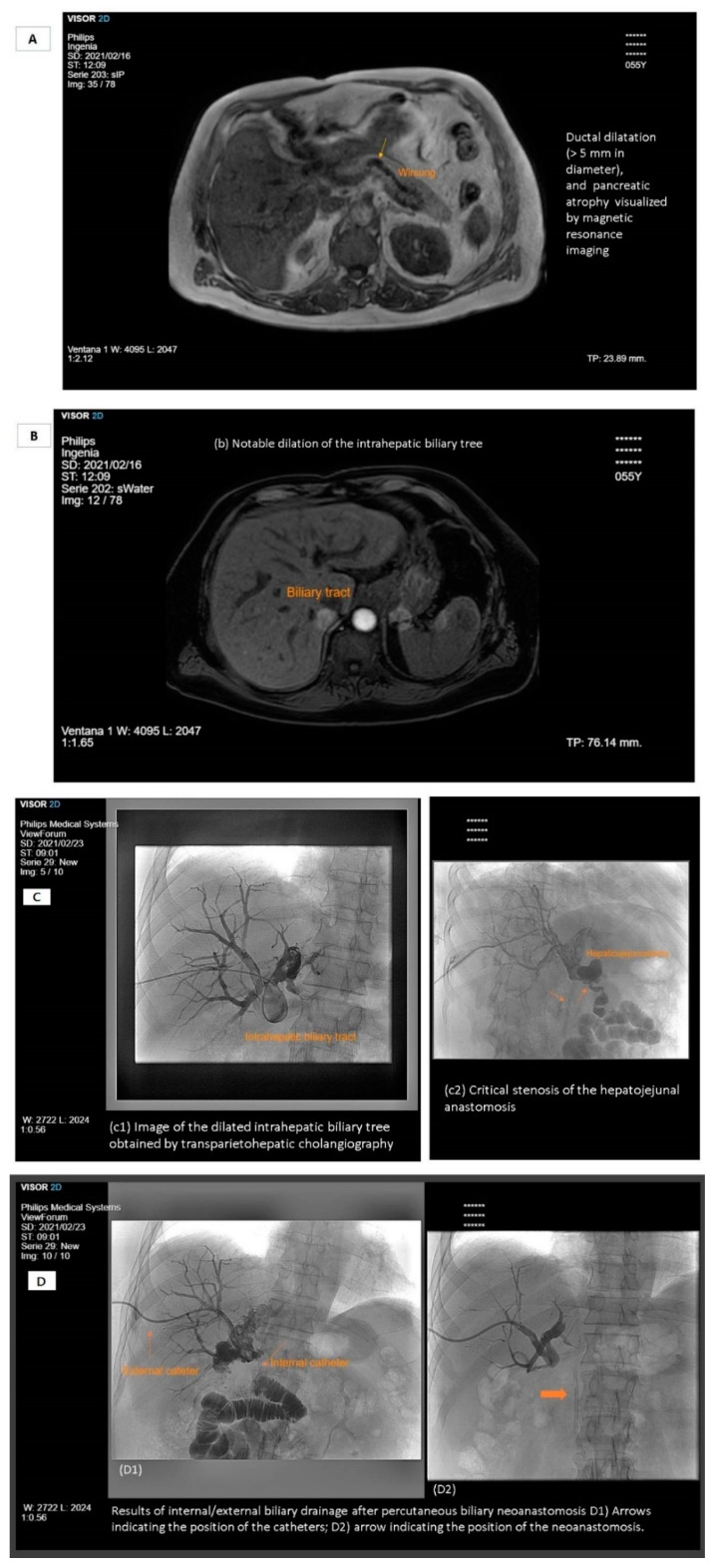
A 55-year-old man was diagnosed with alcoholic chronic pancreatitis 5 years ago, and he underwent regular endoscopic biliary stenting for choledochal stenosis for 1 year. Finally, his choledochal stenosis worsened, and as a result, he underwent a hepaticojejunostomy. For 3 years after surgery, the patient reported jaundice, coluria, acholic stools, steatorrhea, weight loss, and laboratory tests compatible with marked cholestasis, a marked decrease in serum levels of vitamin D (10 ng/mL), and a lengthening of the prothrombin time (16 s). To avoid intraoperative bleeding caused by the development of collateral veins, internal–external drainage was performed by transparietohepatic cholangiography. (**A**) Ductal dilatation (>5 mm in diameter), and atrophy pancreatic visualized by magnetic resonance imaging; (**B**) notable dilatation of intrahepatic ducts due to stenosis of the biliodigestive anastomosis; (**C**) image of the dilated intrahepatic biliary tree obtained by transparietohepatic cholangiography; (**D**) results of internal–external biliary drainage after percutaneous biliary neoanastomosis, (**D1**) arrows indicating the position of the catheters; (**D2**) arrow indicating the position of the neoanastomosis. The patient did not develop postoperative complications, and he was discharged from the hospital with a notable improvement in laboratory tests, appetite, and weight gain.

**Figure 2 nutrients-13-01254-f002:**
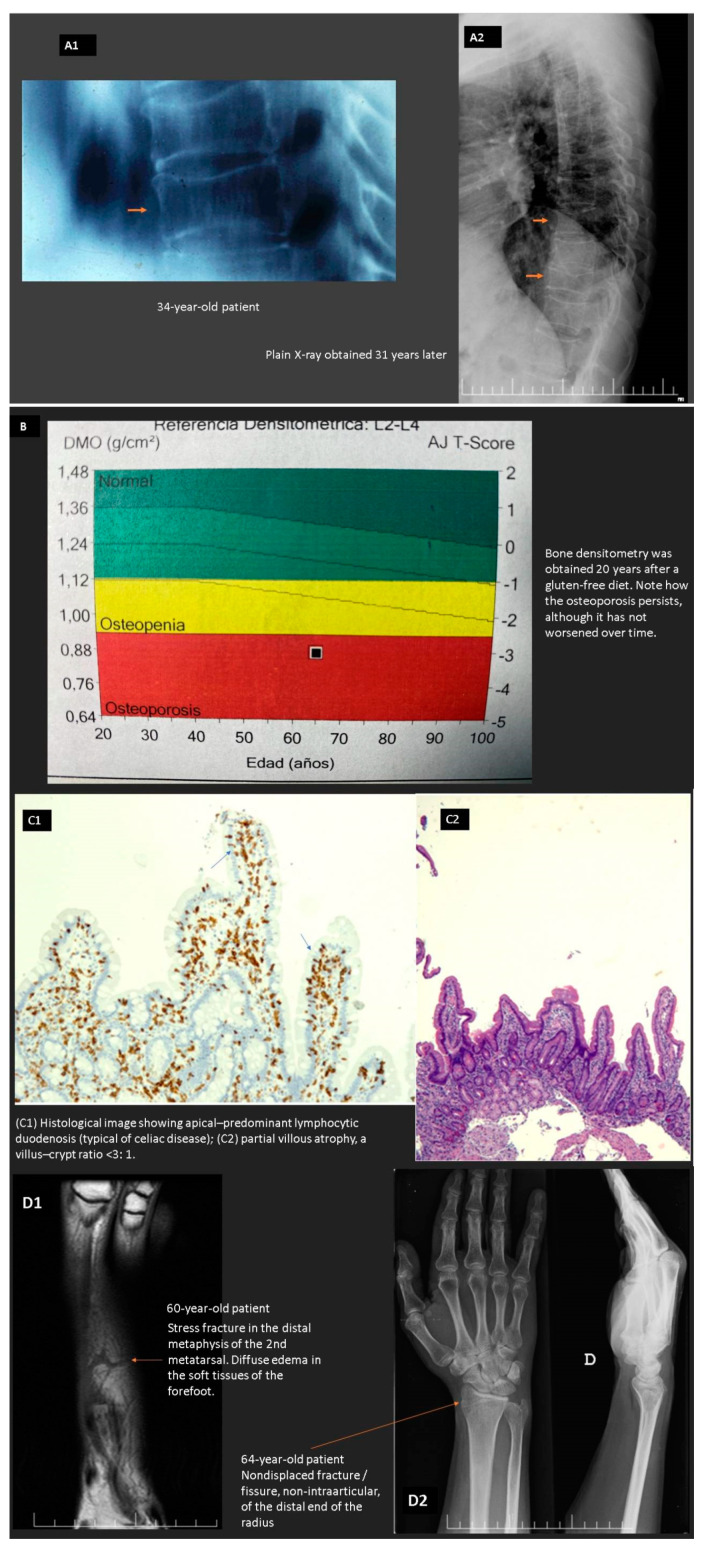
A 34-year-old man suffered a fracture of the 10th dorsal vertebra (**A1**) after an accidental fall. Plain radiography (**A2**) showed signs suggestive of generalized bone demineralization that were later confirmed by bone densitometry (−2.9% in the lumbar spine, adjusted for age) (**B**). After a thorough investigation, he was diagnosed with idiopathic osteoporosis. The patient reported notable growth retardation in childhood, regaining weight, and height in adolescence (adult height 174 cm). The patient reported postprandial distress-type dyspepsia as the only digestive symptom. Nine years later, he underwent a jejunum biopsy ordered by a rheumatologist with experience in idiopathic young adult osteoporosis. The biopsy showed partial villous atrophy, a villus–crypt ratio <3:1, and duodenal lymphocytosis (Marsh–Oberhuber 3a) (**C1**,**C2**). Subsequently, a positive genetic test (HLA-DQ2.5) was confirmed. The case was considered a seronegative celiac disease (non-classical pattern). He is currently 65 years old, and throughout his medical biography he has presented with some other fractures (**D1**,**D2**). In addition to a gluten-free diet, he receives calcium and vitamin D supplements as well as bisphosphonates orally.

**Figure 3 nutrients-13-01254-f003:**
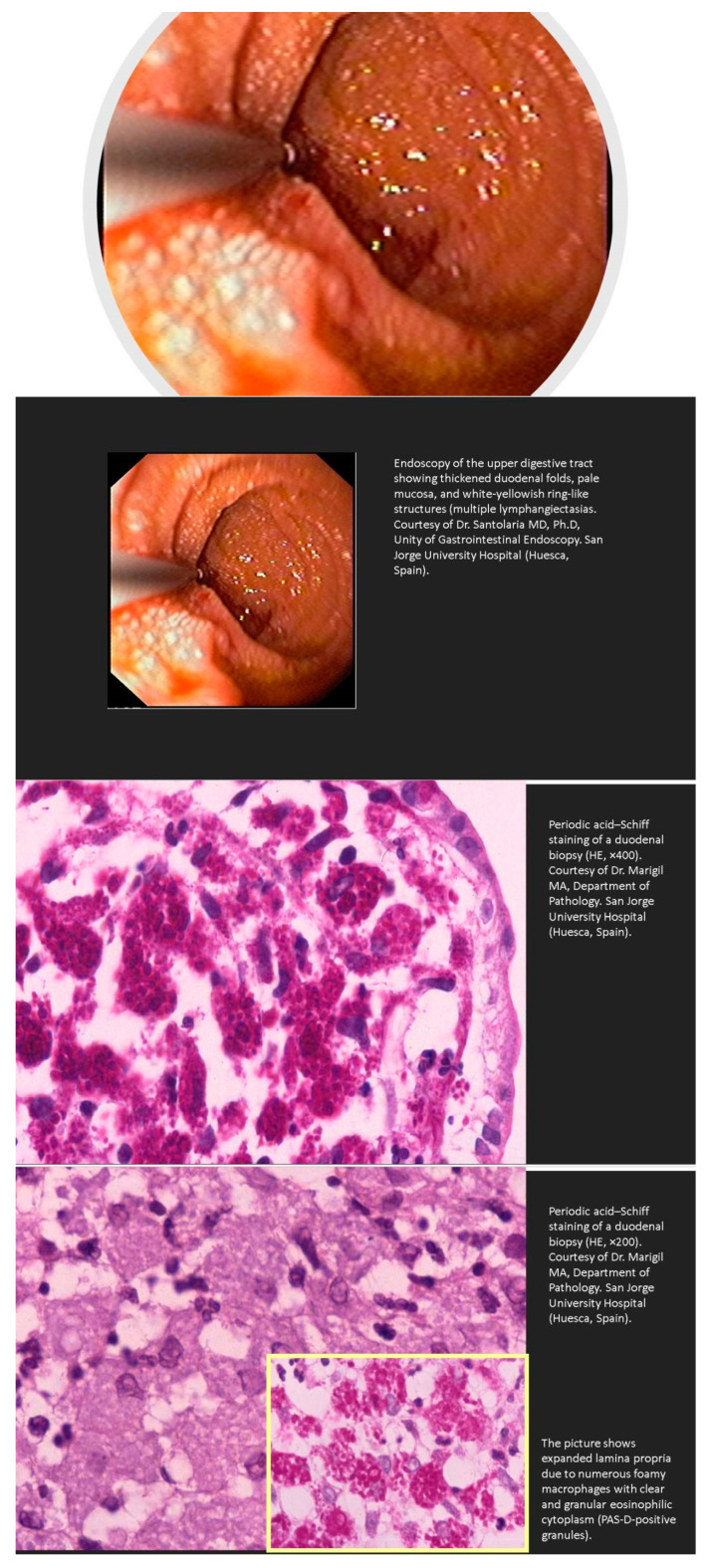
(Whipple disease) The figures correspond to the case of a patient evaluated in our unit more than 20 years ago who presented with abdominal pain, malabsorption syndrome, diarrhea, marked weight loss, arthralgias, and lymphadenopathy, as well as neurological symptoms. A digestive endoscopy was performed, revealing thickened duodenal folds, pale mucosa, and white-yellowish ring-like structures (multiple lymphangiectasias) (Figure 1). The biopsies revealed dilated villi with expanded lamina propria due to numerous foamy macrophages with clear and granular eosinophilic cytoplasm (PAS-D-positive granules). The patient was initially treated with intravenous ceftriaxone, followed by oral trimethoprim/sulfamethoxazole, which led to a significant clinical improvement.

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
