# Peer review of "Small and Large Intestine (I): Malabsorption of Nutrients"

_nutrients, 2021, doi:10.3390/nu13041254_

Round 1

Reviewer 1 Report

An interesting review and comprehesive supported by relevant data and case study.

The introduction needs to highlight the scope of the present review, please improve this section.

However, a conclusion section is missing, and it is needed to conclude the aim and state-of-the-art of the topic discussed.

Author Response

Thank you so much for the comments!

Comment # 1

The introduction needs to highlight the scope of the present review; please improve this section.

Author's response. Please see the next paragraph at the end of the introduction.

"This article will first describe the mechanisms that lead to poor assimilation of nutrients, and secondly, discuss the symptoms and nutritional consequences of each specific disorder. Due to its global vision, this manuscript seeks to be useful both to “registered dietitian-nutritionists” (RDN) and to family doctors, gastroenterologists, internists, and surgeons. Regarding the first (RDN), physicians need to have a reliable ally for their patients' nutritional management, but this is only possible if the RDN demonstrates robust knowledge about the causes and consequences of diseases on nutritional status".

Comment # 2

However, a conclusion section is missing, and it is needed to conclude the aim and state-of-the-art of the topic discussed.

Author's response. Please see the next paragraph at the end of the manuscript:

Conclusion

The normal uptake of nutrients, vitamins, and minerals by the gastrointestinal tract (GI) requires several steps, each of which can be compromised in disease. In healthy conditions, the GI tract will use nutrients, provide energy, and release wastes. However, numerous disorders can alter the physiological mechanisms that guarantee proper digestion and absorption of nutrients (macro-and micro-nutrients), leading to a wide variety of symptoms and nutritional consequences. Knowing the mechanisms that lead to poor assimilation of nutrients, as well as the symptoms and nutritional consequences of each specific disorder, adds value to the role of a wide group of professionals who participate in its management, including RDN, family doctors, internists, gastroenterologists, and surgeons.

Reviewer 2 Report

This is literary review about nutrients malabsorption and the several diseases related to malabsorption etiology and mechanisms. I find the review quite complete, interesting and easy to read, and it is  well illustrated with brief clinical cases. Bibliography is general but acceptably updated.

My only suggestion is to include "olmesartan-associated enteropathy" inside "medication-related" section.

Author Response

Thanks for the feedback and comments.

Comment # 1

My only suggestion is to include "olmesartan-associated enteropathy" inside the "medication-related" section.

Author's Response

We have proceeded as suggested by the reviewer. /